# MetaFetcheR: An R Package for Complete Mapping of Small-Compound Data

**DOI:** 10.3390/metabo11110743

**Published:** 2021-10-28

**Authors:** Sara A. Yones, Rajmund Csombordi, Jan Komorowski, Klev Diamanti

**Affiliations:** 1Department of Cellular and Molecular Biology, Uppsala University, 751 24 Uppsala, Sweden; rajmund.csombordi@gmail.com (R.C.); jan.komorowski@icm.uu.se (J.K.); 2Institute of Computer Science, Polish Academy of Sciences, 01-248 Warsaw, Poland; 3Washington National Primate Research Center, Seattle, WA 98121, USA; 4Swedish Collegium for Advanced Study, 752 38 Uppsala, Sweden; 5Department of Immunology, Genetics and Pathology, Uppsala University, 751 85 Uppsala, Sweden

**Keywords:** small-compound databases, metabolomics, metabolites, queue-based algorithm

## Abstract

Small-compound databases contain a large amount of information for metabolites and metabolic pathways. However, the plethora of such databases and the redundancy of their information lead to major issues with analysis and standardization. A lack of preventive establishment of means of data access at the infant stages of a project might lead to mislabelled compounds, reduced statistical power, and large delays in delivery of results. We developed MetaFetcheR, an open-source R package that links metabolite data from several small-compound databases, resolves inconsistencies, and covers a variety of use-cases of data fetching. We showed that the performance of MetaFetcheR was superior to existing approaches and databases by benchmarking the performance of the algorithm in three independent case studies based on two published datasets.

## 1. Introduction

Metabolomics allows the study of small-molecule substrates and compounds that are involved in metabolic processes. A small compound (<1500 Da) is a low-molecular-weight organic compound that is involved in or may regulate biological processes. Examples of small compounds include various sugars, lipids, and amino acids. Various complex diseases have been strongly linked to metabolic disorders, such as type 2 diabetes and cancer, making metabolomics a highly relevant field for single- and multi-omics studies [1,2,3]. Pathway enrichment analysis is a widespread analysis approach for metabolomics that requires metabolites to map a predefined set of unique identifiers [4]. In this setup there are several issues that arise when accessing, pre-processing, and analysing metabolite data. For instance, the overlapping and non-overlapping information for metabolites is scattered across several small-compound databases, leading to major analysis and standardization issues [5,6,7]. Additional challenges occur with databases that deliver data, which contain multiple entries for one metabolite or incomplete data. Finally, foreign reference identifiers may be missing, making it difficult, sometimes impossible, to find the link between two records of the same metabolite in different databases, while in other cases, the small fraction of reference identifiers that are present might lead to incorrect compounds. The aforementioned issues delay the delivery of results and more importantly, might lead to inconsistent or biased results.

Xia and colleagues developed MetaboAnalyst, which is a versatile computational tool for metabolomics. This tool contains a module aimed at mapping names to identifiers of metabolites from the human metabolome database (HMDB), the chemical entities of biological interest (ChEBI), the Kyoto encyclopedia of genes and genomes (KEGG), PubChem, and METLIN [5,7,8,9,10,11]. However, the lack of a shared nomenclature for metabolite names commonly leads to numerous mismatches or no-matches. Additionally, Wishart and colleagues mapped compounds of HMDB to identifiers of other databases that suffer from inconsistent matches [5]. Moreover, the aforementioned tools map metabolite names to entries in HMDB that may lead to loss of information in case of a mismatch or absence of the metabolite from this specific database.

MetaFetcheR is a unified package targeted towards the metabolomics community, and it is able to resolve multiple inconsistencies and incompleteness in data fetching. The algorithm exhaustively resolves such inconsistent cases and leads to improved mapping of small compounds to identifiers. This is showcased in two case studies using two published datasets by Diamanti et al. and Priolo et al. [1,12] and three existing mappers MS_targeted, MetaboAnalystR along with the webtool MetaboAnalyst 5.0, and Chemical Translation Service (CTS) [1,13,14,15].

## 2. Results

MetaFetcheR is an R package that uses the sparse input of primary database identifiers as a reference point to retrieve identifiers from other databases. The output of MetaFetcheR can be directly incorporated in analysis pipelines. The package unifies data from five open-access and widely used small-compound databases that include HMDB, ChEBI, PubChem, KEGG, and Lipidomics gateway (LIPID MAPS) [16]. Each database has a standardized representation of the identifiers of compounds. The two most widely used representations that are also supported by MetaFetcheR include the simplified molecular input line entry system (SMILES) [17] and the IUPAC international chemical identifier (InChI) [18,19] that describe chemical structures using ASCII characters.

The foundation of the underlying algorithm relies on constructing a local PostgreSQL database that acts as a cache memory of information. Initially, database dump files provided by HMDB, ChEBI, and LIPID MAPS are downloaded. Subsequently, the bulk insertion function of MetaFetcheR is invoked to construct a local database (Appendix A). In the interest of storage space and time, we chose MetaFetcheR to cache data through HTTP calls on the fly from KEGG and PubChem. Cached instances from KEGG and PubChem are stored in the PostgreSQL database for later use in order to avoid unnecessary HTTP calls and timeouts due to excessive calls. A schematic of the local database is illustrated in Figure 1 under the label “Data Repositories”.

The algorithm takes as input a sparse table of known identifiers of a collection of small compounds from the five databases (HMDB, ChEBI, KEGG, LIPID MAPS, PubChem) [5,7,9,16,20] and works on mapping them to identifiers of other databases by filling in the empty fields. This is orchestrated via a queue-based algorithm. Database identifiers are stored in the columns while rows represent metabolites of interest whose identifiers require mapping (Appendix A) (Figure 1-Input table). For each row in the input table, the algorithm appends the known identifiers one by one to a queue (Figure 1-MetaFetcheR main Flow-step 1). The algorithm fetches the db_id, where db is the database and id the identifier, from the top of the queue and queries the respective database for the record with this db_id as a primary identifier (Figure 1-MetaFetcheR main flow-step 2). The algorithm then fills all remaining identifiers for this metabolite with the returned results of the query (Figure 1-MetaFetcheR main flow-step 3). The query return might be empty, in which case, the algorithm issues a new mapping attempt and queries the database again with the db_id as a secondary identifier, which several databases have (Figure 1-MetaFetcheR main flow-step 2). When the queue is empty and the algorithm has filled in most of the empty fields with the newly discovered identifiers (Figure 1-MetaFetcheR main flow-step 3), it reiterates to check whether there are any remaining empty fields (Figure 1-Check Point). In the case of an empty field, a reverse query is issued with one of the identifiers that were resolved during the first pass (Figure 1-MetaFetcheR exceptional flow). The reverse query uses the discovered identifiers to query the respective database in an attempt to fill in the missing field (Figure 1-MetaFetcheR exceptional flow). The algorithm reiterates until all identifiers have been filled in or cannot be further resolved (Figure 2-MetaFetcheR main flow). At the same time, it tracks already discovered records to avoid re-adding them to the queue. Linked identifiers are updated in the local database to avoid future queries and remapping. Moreover, the algorithm stores multiple mapped identifiers of the same compound from the same database that marks ambiguous situations. This allows the user to choose among the discovered identifiers those that are the most fit for downstream analysis. Additionally, it tracks all the identifiers that were used for mapping but returned no result along with the identifiers that were used as secondary_ids and returns the result to the user (Appendix A).

The ability of the algorithm to exhaustively reiterate to resolve cases along with storing multiple discovered database identifiers for the same compound is one of the traits that allows MetaFetcheR to stand out. The algorithm optimizes speed performance by keeping track of formerly discovered records to avoid unnecessary iterations. The size of the database after installation of HMDB, ChEBI, and LIPID MAPS was 189 MB, and after retrieving records for 100 metabolites from PubChem and KEGG, the total size increased by 0.94 MB for the former and 0.20 MB for the latter.

### Usage Scenarios and Benchmarking

MetaFetcheR resolves problematic situations that arise when mapping identifiers of metabolites (Appendix A). We benchmarked the matching performance of MetaFetcheR against three existing mappers in three case studies. We compared the performance of MetaFetcheR to the MetaboAnalystR R package version 3.0.3 using two datasets from Diamanti et al. and Priolo et al. [1,12]. The average mapping rate of MetaFetcheR was ~81% (non-empty fields), while MetaboAnalystR achieved an ~48% average mapping rate on Diamanti et al.’s [1] dataset (Figure 2A-Appendix A). For the dataset from Priolo et al. [12], MetaFetcheR achieved ~95% average non-empty fields rate, while MetaboAnalystR resulted in an ~73% average mapping rate (Figure 2B-Appendix A). Furthermore, we compared the mapping performance of MetaFetcheR to the one of MetaboAnalyst 5.0 web tool and for both datasets, MetaFetacheR performed better (Figure 2B,C, Appendix A). Specifically, the MetaboAnalyst web tool achieved average ~54% mapping rate on Diamanti et al.’s [1] dataset (Figure 2C, Appendix A) and ~78% average mapping rate for the dataset from Priolo et al. [12] (Figure 2D, Appendix A). A similar performance to MetaboAnalyst was observed in the test utilizing CTS and MS_targeted (Figure 3); however, the setting of the experiment was different than that with MetaboAnalystR since the latter can only perform mapping using metabolite names as an input but not metabolite identifiers, while CTS can map using multiple types of metabolite identifiers as input. The mapping rate of MetaFetcheR was on average ~(70%, 68%) (non-empty fields), while CTS achieved ~(38%, 61%) average mapping rate on Diamanti et al.’s [1] and Priolo et al.’s [12] datasets, respectively (Appendix A). For MS_targeted, the average mapping rate was ~34% compared to MetaFetcheR’s ~71% on Diamanti et al.’s [1] dataset (Appendix A). More details about the mapping rate of the three tools compared to MetaFetcheR can be found in Supplementary Note—Results of benchmarking mapping performance of MetaFetcheR. In terms of the running time MetaFetcheR outperformed CTS on both datasets. However, MetaboAnalystR had slightly better running times (Table 1).

In addition to its competitiveness in mapping the identifiers of metabolites, MetaFetcheR also provides insights into the quality of small-compound databases. A test was run by selecting 1000 random identifiers from one of the five databases as input to MetaFetcheR and then we investigated the quality of the collection of retrieved identifiers. The test was performed 100 times for each database (Appendix A). The quality of the databases was assessed using three different metrics: (i) percentage of consistency, (ii) percentage of ambiguity, and (iii) percentage of unresolved cases. Consistency represents the percentage of one-to-one associated cases across all identifiers. Ambiguity is the percentage of original metabolite identifiers linked to multiple identifiers from other databases. Unresolved cases represent the percentage of cases that the original metabolite identifiers failed to link or were absent in all other databases. KEGG showed the highest consistency percentage (~65%) and the lowest fraction of unresolved cases (~23%) compared to HMDB, which had highest fraction of unresolved cases (~71%) (Figure 4).

## 3. Discussion

In this study, we presented MetaFetcheR, an R package for mapping metabolite identifiers across five small-compound databases. Using two published datasets by Diamanti et al. and Priolo et al. [1,12], MetaFetcheR was shown to outperform other existing tools, such as MS_targeted, MetaboAnalyst (both R package and webtool v5.0), and CTS, that provide similar mapping functionalities. Additionally, MetaFetcheR showed a reasonable running time compared to similar tools even given its exhaustive reiteration. For instance, MetaboAnalystR and MetaboAnalyst 5.0 web tool were slightly faster, which indicates that the better mapping performance of MetaFetcheR compared to MetaboAnalystR and MetaboAnalyst 5.0 web tool was slightly compromised by the running time. MetaFetcheR will be continuously updated to include additional databases. Features that include searching by metabolite name and automatic updates of the local database will be provided in future builds. A limitation of our current solution is that the user might be required to manually curate the output dataset in order to resolve the inconsistencies of the databases.

## 4. Materials and Methods

MetaFetcheR package was developed using R version 3.5. All the database queries for installation and mapping tasks are performed using PostgreSQL version 12. PostgreSQL is a free and open-source database and is released under the PostgreSQL License, which is a liberal Open-Source license.

MetaFetcheR is available at https://github.com/komorowskilab/MetaFetcheR/ (accessed on 17 October 2021). Details about installing the package, the prerequisites, and the main functions for running MetaFetcheR are available at https://komorowskilab.github.io/metafetcher/ (accessed on 14 October 2021).

The SQL dumps were downloaded to install the local database for testing on 7 May 2020.

In order to allow reproducibility of the results, we placed all the scripts used to generate the results and figures in https://github.com/komorowskilab/MetaFetcheR_Experiments (accessed on 22 October 2021).

### 4.1. Performance Measures

Mapping Rate_db_id_ represents the coverage of each type of database identifier after running tool X.

db_id represents the type of identifier (HMDB, ChEBI, LIPID MAPS, KEGG, or PubChem). The number of non-empty fields_db-id_ (X) represents the number of non-empty fields for the db_id identifier type after running tool X. The total number of records represents the total number of metabolites records in the input table:(1)$${\mathrm{Mapping}\;\mathrm{Rate}}_{\mathrm{db}\_\mathrm{id}}\left(X\right)=\frac{\mathrm{Number}\;\mathrm{of}\;{\text{non-empty}\;\mathrm{fields}}_{\mathrm{db}\_\mathrm{id}}\left(X\right)}{\mathrm{Total}\;\mathrm{number}\;\mathrm{of}\;\mathrm{records}}$$
(2)$${\mathrm{Average}\;\mathrm{Mapping}\;\mathrm{Rate}}_{\left(i_{1},\;i_{2},\;i_{3},\;\ldots ,\;i_{k}\right)}\left(X\right)=\frac{{\mathrm{Mapping}\;\mathrm{Rate}}_{i_{1}}\left(X\right)+{\mathrm{Mapping}\;\mathrm{Rate}}_{i_{1}}\left(X\right)+\cdots {\mathrm{Mapping}\;\mathrm{Rate}}_{i_{k}}\left(X\right)}{k}$$

The average mapping rate is the summation of all mapping rates of tool X on all identifier types (i_1_, i_2_,..., i_k_) divided by the number of identifier types (k):(3)$${\mathrm{Percentage}\;\mathrm{of}\;\mathrm{unmapped}}_{\mathrm{db}\_\mathrm{id}}\left(X\right)=\frac{{\mathrm{Number}\;\mathrm{of}\;\mathrm{empty}\;\mathrm{fields}}_{\mathrm{db}\_\mathrm{id}}\left(X\right)}{\mathrm{Total}\;\mathrm{number}\;\mathrm{of}\;\mathrm{records}}$$
where percentage of unmapped_db_id_ (X) represents the percentage of empty fields for each identifier type after running tool X. db_id represents the type of identifier (HMDB, ChEBI, LIPID MAPS, KEGG or PubChem). The number of empty fields_db_id_ (X) represents the number of empty fields for the db_id identifier type after running tool X. The total number of records represents the total number of metabolites records in the input table:(4)$${\mathrm{Matching}\;\mathrm{percentage}}_{\mathrm{db}-\mathrm{id}}\;\left(\mathrm{MetaFetcheR},\;Y\right)=\frac{{\mathrm{Number}\;\mathrm{of}\;\mathrm{matching}\;\mathrm{feilds}}_{\mathrm{db}\_\mathrm{id}}\left(\mathrm{MetaFetcheR},\;Y\right)\;}{{\text{non-empty}\;\mathrm{fields}}_{\mathrm{db}\_\mathrm{id}}\left(Y\right)}$$
where matching percentage_db_id_ (MetaFetcheR, Y) represents the percentage of matching identifiers between both tools MetaFetcheR and Y after running them for the identifier type db_id. db_id represents the type of identifier (HMDB, ChEBI, LIPID MAPS, KEGG, or PubChem). The number of matching fields_db_id_ (MetaFetcheR, Y) represents the number of matching identifiers between both tools MetaFetcheR and Y after running them for the identifier type db_id. Non-empty fields_db_id_ (Y) represents the number of non-empty fields for db_id identifier type after running tool Y.

### 4.2. Benchmarking Mapping Performance of MetaFetcheR

The performance of MetaFetcheR was benchmarked based on three case studies using two datasets and three existing tools, with the first dataset by Diamanti et al. [1] and the second one by Priolo et al. [12]. The three tools are MS_targeted, MetaboAnalystR along with MetaboAnalyst 5.0 web tool, and Chemical Translation Service (CTS) [1,13,14,15].

#### 4.2.1. Case 1

We compared the performance of the algorithm for mapping metabolite identifiers to the identifiers mapped by MS_targeted on Diamanti et al.’s [1] dataset. MS_targeted is a command-line tool that was used earlier for mapping metabolites’ identifiers. The comparison was based on the rate of mapped and unmapped metabolite identifiers from both tools. Subsequently, the results from MS_targeted were manually curated and the concordance between MetaFetcheR and the manual curation of MS_targeted results was assessed.

#### 4.2.2. Case 2

We compared the MetaFetcheR mapping performance to that of the compound ID conversion function of MetaboAnalystR [13] and MetaboAnalyst 5.0 webtool using data from [1,12]. MetaboAnalystR accept input in the form of metabolites’ names, which is problematic since there is no commonly accepted nomenclature across small-compound databases. Both MetaboAnalystR and MetaboAnalyst5.0 cannot map to LIPID MAPS identifiers. Based on these limitations, the comparison of the mapping performance with MetaboAnalystR was restricted solely to the metabolites that were mappable and was limited for both tools to HMDB, ChEBI, KEGG, and PubChem identifiers. The list of metabolites that could not be mapped is shown in Appendix A. The comparison of the mapping performance was based on the rate of mapped and unmapped metabolite identifiers when using metabolite names as input for both MetaboAnalyst tools. Unlike MetaboAnalystR, MetaboAnalyst 5.0 accepts metabolite identifiers as input. In addition to the previous comparison, we also compared the number of metabolite identifiers that MetaboAnalyst 5.0 webtool mapped when the input was the available HMDB, KEGG, and LIPID MAPS identifiers in Diamanti et al.’s [1] dataset, and the available KEGG identifiers in Priolo et al.’s [12] dataset.

#### 4.2.3. Case 3

We compared the MetaFetcheR mapping performance to that of CTS using data from [1,12]. CTS accepts lists of metabolites’ names or metabolite identifiers of the same kind as input. Additionally, CTS does not support PubChem identifiers. To achieve a fair comparison, we ran CTS and MetaFetcheR three times with the available HMDB, KEGG, and LIPID MAPS identifiers in Diamanti et al.’s [1] dataset and the available KEGG identifiers in Priolo et al.’s [12] dataset. For instance, the input table to MetaFetcheR in case of using the available HMDB identifiers in Diamanti et al.’s [1] dataset had a complete HMDB_id column and the rest of the identifiers’ columns were empty whereas the input to CTS was the same list of HMDB identifiers. ChEBI identifiers were discarded from the comparison using Diamanti et al. [1] since there were only two available entries.

Figure 1, Appendix A were created using www.lucidchart.com (accessed on 14 October 2021) resources. Appendix A was created using www.cacoo.com (accessed on 14 October 2021) resources.

## Figures and Tables

**Figure 1 metabolites-11-00743-f001:**
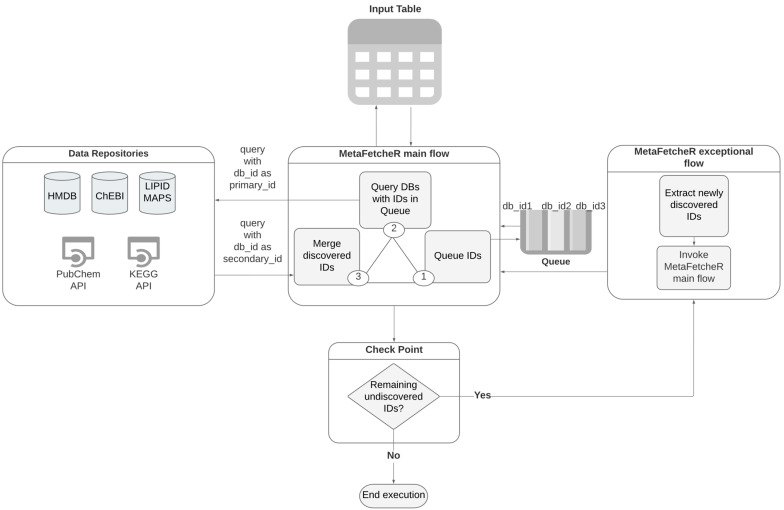
A simplified graphic illustration of the MetaFetcheR algorithm. Data repositories represent the local database built during installation of the package and the http calls to the application programming interfaces (APIs) for PubChem and KEGG. MetaFetcheR main flow represents the main flow of the algorithm, which constitutes three main steps. Queue represents the working queue data structure the algorithm utilizes to add all the primary and secondary known IDs from the input table and consequently preform the search queries with the IDs present in the queue. Check point is the step when the algorithm has emptied the queue after a round of search to check if the input table still has empty fields, in which case, it utilizes the steps in MetaFetcheR exceptional flow. A detailed version is available in Appendix A.

**Figure 2 metabolites-11-00743-f002:**
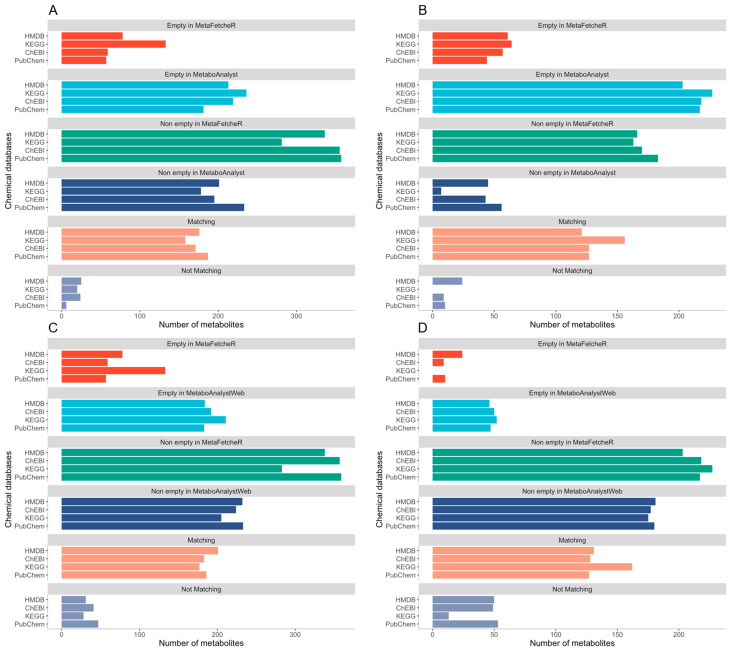
Comparison of the mapping performance of MetaFetcheR to MetaboAnalystR and MetaboAnalyst 5.0 webtool. (**A**) Comparison of the mapping performance for MetaFetcheR and MetaboAnalystR on Diamanti et al.’s [1] dataset, (**B**) Comparison of the mapping performance for MetaFetcheR and MetaboAnalystR on Priolo et al.’s [12] dataset, (**C**) Comparison of the mapping performance for MetaFetcheR and MetaboAnalyst 5.0 webtool on Diamanti et al.’s [1] dataset, (**D**) Comparison of the mapping performance for MetaFetcheR and MetaboAnalyst 5.0 webtool on Priolo et al.’s [12] dataset. Empty in the MetaFetcheR and MetaboAnalyst or MetaboAnalystWeb panels illustrates the number of identifiers that could not be mapped using the respective tool. Non-empty in the MetaFetcheR and MetaboAnalyst or MetaboAnalystWeb panels presents the number of identifiers that were successfully mapped using the respective tool. Matching panel shows the number of mapped identifiers that agreed between tools. Non-matching panel shows the number of mapped identifiers that were not in agreement between tools. The number of identifiers is shown on the x-axis.

**Figure 3 metabolites-11-00743-f003:**
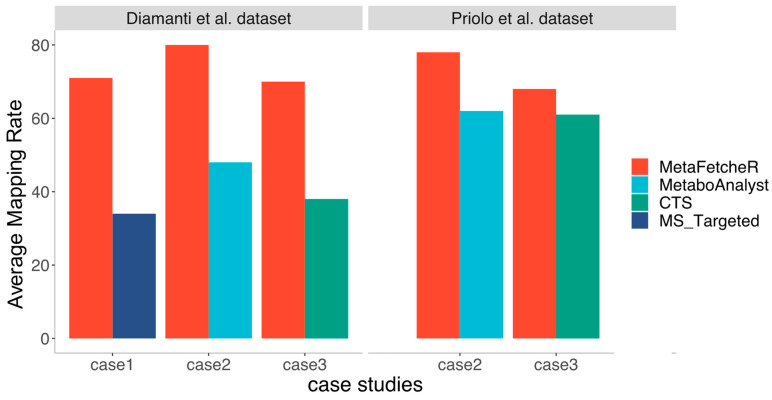
Comparison of the mapping performance for the three tools with the mapping performance of MetaFetcheR. The comparison utilizes the data from the three case studies performed on the two datasets from Diamanti et al. [1] and Priolo et al. [12].

**Figure 4 metabolites-11-00743-f004:**
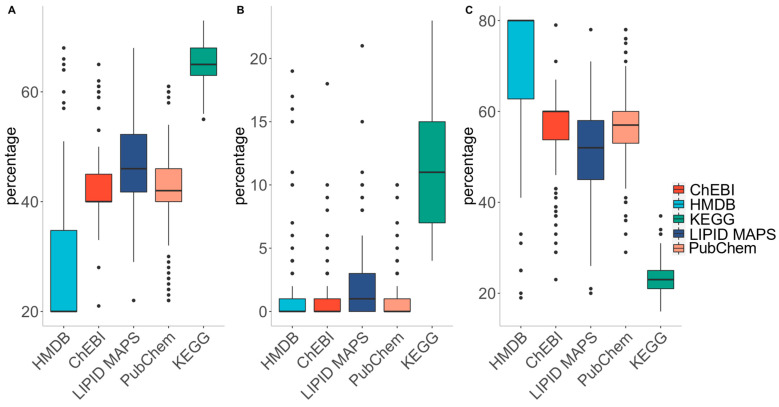
The results for the test that was run on each database to investigate data quality. (**A**) Box plot for the fraction of successfully mapped identifiers that had one-to-one mappings to the rest of the database metabolite identifiers after running the test 100 times for each database. (**B**) Box plot for the fraction of successfully mapped identifiers that had at least one one-to-many mappings to the rest of the database metabolite identifiers after running the test 100 times for each database. (**C**) Box plot for the extent of unmapped identifiers to the rest of the database metabolite identifiers after running the test 100 times for each database.

**Table 1 metabolites-11-00743-t001:** Comparison of the running time for MetaFetcheR, MetaboAnalystR, and CTS.

Tool	Number of Input Metabolites	Input Type	Dataset	Running Time
Metaboanalystr	434	Metabolites names	Diamanti et al. [1]	1 min
	228	Metabolites names	Priolo et al. [12]	30 s
Metafetcher	434	HMDB, ChEBI, LIPID MAPS, KEGG, PubChem identifiers	Diamanti et al. [1]	5 min
	228	HMDB, ChEBI, LIPID MAPS, KEGG, PubChem identifiers	Priolo et al. [12]	2 min
	328	HMDB identifiers	Diamanti et al. [1]	1 min
	219	KEGG identifiers	Diamanti et al. [1]	1 min
	68	LIPID MAPS identifiers	Diamanti et al. [1]	20 s
	228	KEGG identifiers	Priolo et al. [12]	2 min
CTS	328	HMDB identifiers	Diamanti et al. [1]	4 min
	219	KEGG identifiers	Diamanti et al. [1]	10 min
	68	LIPID MAPS identifiers	Diamanti et al. [1]	4 min
	228	KEGG identifiers	Priolo et al. [12]	18 min

## Data Availability

Data is contained within the article or Appendix A. The data presented in this study are available in Diamanti et al. [1] and Priolo et al. [12].

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
