# Peer review of "MetaFetcheR: An R Package for Complete Mapping of Small-Compound Data"

_metabolites, 2021, doi:10.3390/metabo11110743_

Round 1
Reviewer 1 Report
In their work Yones et al. describe a R Package to map different Metabolite IDs. In fact Metabolomics suffers even more that Genomics from non-unique, ambiguous and inconsistent metabolite identifiers. Thus therir approach to develop a tool to cross-map different Metabolite IDs is of great intertest to the community.
Unforuntaltely, their approach, first and foremost its presentation is of poor quality. The whole manuscript suffers from poor written english and an inaccurate and sloppy (sorry) description.
It is understandable that the tool fills up a table with different IDs from different databases, however it is not clear how the tool deals with the major difficulties for this task: many to many mappings, non-unique mappings, wrong mappings, inconsistencies etc...
Comparison of the results is only partial informative. how about false positive rates, accuracy, missing data rates ...
Figures are of poor quality, labels often not readable or missing
However this is an important task and usable tools to deal with these problems are needed, I suggest the authors completely revise their method, method evaluation, manuscript and figures.
Reviewer 2 Report
Review of submission
“MetaFetcheR: An R package for complete mapping of small compound data”
The authors are presenting MetaFetcheR, a new tool that fits into the niche of establishing the correct mappings between identifiers of different compound databases used in the metabolomics data-analysis workflow. Is partially solves a problem that seems to be deceptively trivial, since cross-database readouts are simple to implement and include the identifiers. But, for the reasons pointed out by the authors in the introduction and from their results on database consistency, unexpected problems do arise, and this tool is an important contribution at solving them. It is my opinion that this tool will appeal to the wide community of researchers doing metabolomics.
The paper is well written, easy to read, and the use cases are compelling examples that the tool works and seems to be superior to the others developed to achieve the same goal.
There are minor of issues that I believe that the authors should address, to further improve this paper:
- Algorithm inner workings. I understand that the textual description of the algorithm and the supporting diagram in Figure S2 go into such a deep level of detail that their place is in the supplementary materials of the submission. However, a less vague presentation of the algorithm in the main text would, in my opinion, would make the paper more explicit, more convenient for the readers and better support the downstream results/discussion. Figure 1 is vague and almost useless, using concepts that were not properly defined at this point of the paper, like “main flow” and “exceptional flow”. Figure S2 and the technical note shed light into the inner workings of the algorithm. I suggest bringing more detail and information from the technical note into the main text, in the location where the exhaustive search is first mentioned, and incorporate some of the details from Figure S2 into Figure 1. Key ideas like what exactly goes into the queue (first primary and the secondary IDs), and why the search is actually “exhaustive” have their place in the main text.
- Performance definition. I think that a formal definition of mapping performance, presented as percentages throughout the results, is needed at some point (perhaps in the M&Ms section). The reason is that there is some ambiguity in the interpretation of this metric. Is mapping performance the fraction of input IDs for which the exhaustive mapping was successful? The fraction of input IDs for which a total mapping to the other DBs was accomplished? Is it computed on a per-field basis, being the fraction of fields that were nonempty in the end?
- Not one-to-one ID relationships in the DS. If the correspondences between DB Ids are not on-to-one (two cis/trans isomers have the same ID in one DB but not in another one, for example), how is this handled? Are all the possible Ids presented to the user for further “downstream” curation? I suspect that this is the case, but can the authors make it explicit?
- Technical details of the package and installation are missing. Even if MetaFetcheR follows all the standards of CRAN/Bioconductor and is trivial to install for R users, some indication about the documentation should be given in the paper, along the statement whether it is a command line, web front-end or API only tool. Furthermore, details are missing about the bootstrap installation of the DBs. It is automated by the PostgreSQL, sure, but the version or time stamp of the DBs should be indicated somewhere, along with the issues regarding DB licensing.
- The list of authors ends with the word “and”. Most likely to indicate that author KD affiliations and the fact that he is also the corresponding author, but the “and” can be removed?
- The sentence in lines 35-36 in the introductions seem to suggest that the source of problems is the format of information transfer (MySQL dump) and not the structure and content of the DB. I believe that this is not the case.
Reviewer 3 Report
The Authors of the manuscript present an R software for mapping small compound data. The Authors explain in a convenient way the advantages of their software. However, I did not understand what did the Authors mean by “small” compound data. Please clarify this in the text of the manuscript. The Authors also did not mentioned what are the limitations of the software and when one could expect these limitations or possible errors to appear.
Reviewer 4 Report
The manuscript is well written and easy to read. However, it needs a full revision due to the lack of scientific novelty. It does not follow the common order of sections. Materials and methods should be before results. Furthermore, the proposed problem has been discussed and mostly solved already by unichem or CEU Mass Mediator (https://www.ebi.ac.uk/unichem/ucquery/listSources,
https://www.sciencedirect.com/science/article/abs/pii/S0731708517326559). The compound mapping is a necessity in the field, but the usage of a local database is a known solution to improve the efficiency and has already been applied in both solutions previously mentioned. Although the proposed solution differs slightly from both solutions and it may be useful for some users aiming to get information from different sources, it still leave some concerns that should be addressed in the manuscript and improved. They are enumerated below:
1- What is the objective definition of an inconsistent match? Right now there are structures with inconsistent information in the sources. If it has been verified, how was it done? It should be stated in the manuscript. If it was performed manually, datasets of 1,000compounds does not look enough since databases contain usually ~150,000 compounds. If they have taken metrics in some other automatic way about what the authors consider a consistent match, it should be stated in the manuscript.
2- The usage of a local database seems the best alternative to improve the performance for subsequent tasks in metabolomics. However, if the final goal of MetaFectheR is just the mapping of compounds, how can it verify that local information is up to date and it did not change in the sources? The metabolomic databases are continuously updated and growing. In an ideal situation, an identifier should not belong to different structures ever, but the information there stored does change. How often is MetaFetcheR updating its local database? How it can affect to the source databases? The dumps provided by databases are usually performed with a frequency higher than one month, so there is another point of potential outdating of information.
3- How does the queue-based algorithm works? It is not explained in detail. The paragraph describing it is vague and unconcise in lines 83-90.
4- For a potential user, what are the alternatives of compound mapping and what is the difference between MetaFetcheR and the alternatives? What problem does it propose MetaFetcheR that is not solved? In case it is solved, what does it make MetaFetcheR
different? It does not look that any user interested in an ID will benefit from it.
5- The inclusion of a MetaFetcheR will include even more dispersion to the current state of the art. The InChI and its corresponding InChI Key are descriptive enough to be a unique
identifier followed by current databases. Summarizing, the solution here provided only collects a list of identifiers from the entrance of a single one. It does not look enough for a manuscript of a high-level journal. The work performed is a good start point to use in different applications, but the search and unification of compounds from an identifier does not constitute a novelty in the field.
Round 2
Reviewer 1 Report
In this revised version the manuscript "MetaFetcheR: An R package for complete mapping of small compound data" improved. But still I think it need further editing
Quality of written English is still low. Inroduction: ' ... low molecular weight organic compound that is involved or may regulate ...' should be ' low molecular weight organic compound that is involved IN or may regulate...'; Results: 'The local database is represented in Figure 1 as Data Repositories.'; ' When the algorithm has filled in all the possible empty fields in each row that could be mapped using the primary or secondary identifiers that were initially given ...' (wording!)
Figures labels are hard to read, due to small font sizes
Reviewer 4 Report
The manuscript has been properly reviewed and its purpose is now clearer. It describes an algorithm to collect identifiers based on the previous link between databases. This problem is of critical importance when using pathway or biological analysis tools that require a certain type of identifiers and currently the researchers and doing this work manually.
The technical work is great, the algorithm works well and it is proven that it links more identifiers than another solutions using a sequential queue system. Nevertheless, the algorithm relies in local mappings performed individually in each database that are wrong in more cases that we would like to. Isomeric and differences in bond types are usually ignored by database linkings between compounds. Unfortunately, it is not considered the way of solving the problem in the field. The way of linking compounds from different databases should be the chemical structure (InChIs preferably, SMILES, SDFs, mol files, etc..). The problem of manually unifying identifiers from different databases is now transformed in the problem of manually confirming the right linking in the databases. Furthermore, if the database link is wrong (inconsistency), it may lead to wrong biological interpretations performed by the subsequent computational tool in the pipeline. It cannot happen using the chemical structures, and it was actually created to solve this problem. In that case, the problem of unifying compounds and linking identifiers would have been solved by other tools like ChEBI or CEU Mass Mediator. Even though, to have a specific R package to solve the problem of obtaining identifiers from a list of identifiers may be useful, but it has to be addressed using the chemical structure, not relying on external linking of identifiers. What would it happen if a user introduces two identifiers in the same row belonging to two different compounds? These incosistencies are potentially harmful.
The tool can be adapted and use the InChI Keys provided by all databases to unify compounds. Moreover, it can also automate the way of extracting MySQL dumps from databases (LipidMaps, HMDB, ChEBI provide downloads in permanent links and different type of dumps - even PostgreSQL in ChEBI -).
The information about compounds is duplicated. Relational databases are intended to provide consistency and avoid replication of data, while the solution provided here permits the inconsistency (IDs linked with different Formula and chemical structure) and it contains replication of data (compound properties in each database entity).
Figure S1 does not represent an E-R model, it represents just the entities of the database. Every FK is represented as a relationship in an E-R diagram, and no relationships are present in the diagram.
The SQL dumps to install the local database must be updated (May 2020 is the current version).
Round 3
Reviewer 4 Report
The information about compounds is duplicated. Relational databases are intended to provide consistency and avoid replication of data, while the solution provided here permits the inconsistency (IDs linked with different Formula and chemical structure) and it contains replication of data (compound properties in each database entity).
The E-R model in figure S1 is now corrected and correct, but it still shows this replication of data mentioned above. A central entity containing common information shall be used in the next version of the software, if not possible in this one.
The software is useful for users that want identifiers of databases for a subsequent analysis in software that require specific IDs and it is demonstrated that the software provides a faster and more complete solution to the problem presented.